# Personalized Visual Content Generation in Conversational Systems

**Xianquan Wang[1*],   Zhaocheng Du[2*],   Huibo Xu[1*],   Shukang Yin[1],   Yupeng Han[1],
Jieming Zhu[2],   Kai Zhang[1†],   Qi Liu[1]**

[1]University of Science and Technology of China
[2]Huawei Noah's Ark Lab

{xqwang, xhbxhb, xjtupanda, yupenghan}@mail.ustc.edu.cn
zhaochengdu@huawei.com  jiemingzhu@ieee.org
{kkzhang08, qiliuql}@ustc.edu.cn

## Abstract

With the rapid progress of large language models (LLMs) and diffusion models, there has been growing interest in personalized content generation. However, current conversational systems often present the same recommended content to all users, falling into the dilemma of "one-size-fits-all." To break this limitation and boost user engagement, in this paper, we introduce PCG (**P**ersonalized Visual **C**ontent **G**eneration), a unified framework for personalizing item images within conversational systems. We tackle two key bottlenecks: the depth of personalization and the fidelity of generated images. Specially, an LLM-powered *Inclinations Analyzer* is adopted to capture user likes and dislikes from context to construct personalized prompts. Moreover, we design a dual-stage LoRA mechanism—*Global LoRA* for understanding task-specific visual style, and *Local LoRA* for capturing preferred visual elements from conversation history. During training, we introduce the visual content condition method to ensure LoRA learns both historical visual context and maintains fidelity to the original item images. Extensive experiments on benchmark conversational datasets—including objective metrics and GPT-based evaluations—demonstrate that our framework outperforms strong baselines, which highlight its potential to redefine personalization in visual content generation for conversational scenarios like e-commerce and real-world recommendation.

## 1   Introduction

The rise of large language models (LLMs) [34, 35] has sparked conversational systems across industries. For instance, e-commerce platforms can recommend preferred items through multi-turn conversations [30, 5], while music apps interactively suggest album cover [3]. They operate by dynamically analyzing users' inclinations (likes or dislikes) during interactions [14]: when users show strong interest in specific item categories, or strongly advocate for a specific item, the dialog prioritizes related content; conversely, explicit rejections reduce recommendations about those items [12, 21]. Usually, the systems also include the **item image** to attract users, as shown in Figure 1.

*A picture is worth a thousand words*; images are the most intuitive and vivid representation of items. Unfortunately, current conversational systems **presented identical item representations** to all users, even though different users prefer different points of the same item. For example, the

---

*Equal contribution
†Corresponding author: Kai Zhang (`kkzhang08@ustc.edu.cn`)

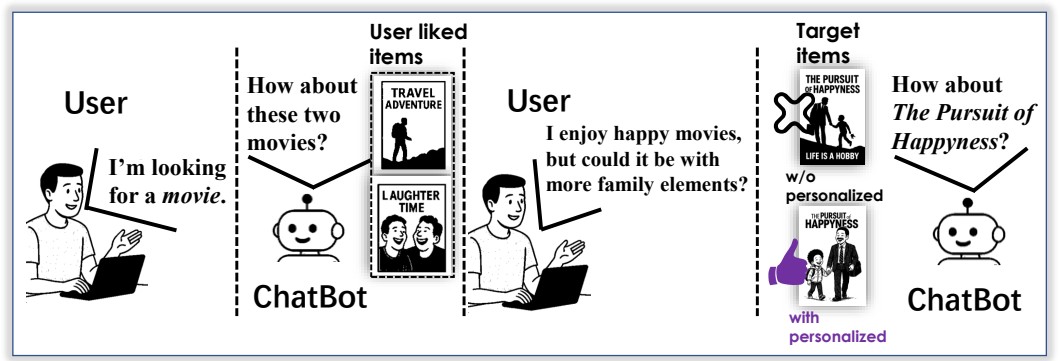

Figure 1: A toy example of content generation personalization in **movie conversational systems**. In this example, the personalization focus on visual content.

parenting, adventure, and inspirational themes of movie *The Pursuit of Happyness* attract different users. Incorporating personalized image content into the system is expected to significantly enhance the system's level of personalization, which is expected to lead to a more satisfying user experience, increase user engagement, and improve alignment between user intent and system output—particularly in application domains such as entertainment [31], e-commerce [18], and digital assistants [37].

However, little to no work has systematically explored personalized visual content generation within conversational systems, and achieving this goal is far from easy. In the conversation process, generating the visual content that reflects the user's preferences on the target item faces three main challenges. First, **how to extract the user's inclinations that can be understood by content generator** based on the conversation content. These inclinations consist of two parts: one is the positive preference, or what is known as "likes"; the other is the aversion to certain styles, or "dislikes." These inclinations may reflect on certain style types or may be deeply hidden in historical images. This requires a thorough understanding [7] and analysis of the conversation history to accurately and effectively summarize the user's inclinations. Another challenge is **how to incorporate the user's preferred visual elements** into the image generation process, as even if a user enjoys "comedy" films, the preferred elements may vary greatly. The image references in conversational systems can provide beneficial guidance. Finally, **how to preserve the identity** in the generation process, ensuring that while personalization is reflected, the identity characteristics (such as the actor or semantic information) remain as consistent as possible during the inclinations' conditioning process.

To address the three challenges mentioned above, we constitutes one of the earliest comprehensive attempts to form a unified framework **PCG** (**P**ersonalized **V**isual **C**ontent **G**eneration) in conversational systems. **First**, we utilize a language model to summarize and analyze past interaction items, extracting both positive and negative user preferences. In this process, the language model acts as a analyzer, with word weight modeling to ensure that the output remains within expectations. These summaries form user preference descriptions, which guide the generation process to include or exclude content. **Second**, integrating previously favored content elements into the generation is also challenging. Based on the contextual understanding ability of base models, we propose a strategy that directly incorporates historical item images by concatenating each other on a unified canvas. This allows the generation process to be conditioned on visual preferences without modifying the model architecture. **Finally**, ensuring identity consistency requires the model to understand the relation between the target item and historical images. To facilitate this, we propose the sequence invariance strategy for historical items condition, and introduce global-local LoRA mechanism for guidance, which help the model better capture this relationship.

For evaluation, we adopt two commonly used benchmark datasets in conversational systems to validate the effectiveness of our method. In the objective experiment, we use various low-level and high-level metrics to assess the degree of personalization and fidelity of the generated outputs. In the GPT evaluation study, we further evaluate image aspects such as style and coherence. Experimental results demonstrate the effectiveness of our approach, and this work may open up a new research direction at the intersection of conversational AI and visual generation. The code is publicly available at https://github.com/xqwustc/PCG.

## 2  Related Work

### 2.1  Conversational Systems

Conversational systems [20] have been widely explored in fields such as robotics, smart shopping assistants, and recommender systems [26, 33, 28]. With the advancement of language models [38, 36] for text [17] and language processing, these systems have evolved from handling simple dialogues to understanding complex semantics [34]. The rise of large language models has enabled deeper comprehension, allowing systems to better understand user needs and empathize with user emotions [13]. This enhances the sense of engagement and satisfaction during interactions [39]. However, most current systems focus only on interaction at each turn, with little attention paid to personalizing rich visual content based on the conversations. Our work addresses this gap by introducing a novel approach to content personalization within conversational systems.

### 2.2  Content Generation

With the rapid development of large language models and visual generation models, personalized content generation has drawn increasing attention [22, 32]. For instance, personalization in large models enables the generation of responses tailored to users' specific needs and preferences. The primary goal of such personalization is to enhance user satisfaction by producing outputs that better align with individual requirements and expectations. A key function of this type of personalization is to generate text content that matches the user's writing style [15, 1]. These approaches have been applied in various contexts, such as user-level personalization based on interaction history, role-level customization for specific personas, and global preference alignment (*e.g.*, ethics, factuality). However, these efforts have been predominantly limited to text-based personalization.

**Personalization of visual content** has also been widely explored. For example, several studies have used Diffusion-based models [6, 24, 4] or the FLUX architecture [27] to customize image generation. Their success is built on the foundation of **Diffusion Transformers (DiTs)**. These methods support prompt-driven generation and style transfer, among other features. Recently, large multimodal models [2, 11] have integrated both understanding [16] and generation capabilities. However, they still lack the ability to actively infer user preferences and rely heavily on explicit user input. As highlighted in pioneering works [25, 29, 35], some studies have attempted to explore personalization in the multimodal domain (*e.g.*, images). Nevertheless, these works fail to model user interests from **conversational history** [8], leaving a blank in personalizing visual content within multi-turn conversations.

## 3  Preliminaries

Let $\mathcal{P}$ be the set of participants and $\mathcal{I}$ the set of items. A *conversation* is represented as a sequence of interaction turns: $C = (p_t, s_t, i_t)_{t=1}^{T}$, where at each turn $t$, a participant $p_t \in \mathcal{P}$ (either a *user* or a *chatbot*) produces an utterance $s_t$, which may mention a subset of items $i_t \subseteq i$. The set $i_t$ can be empty if no items are referenced. The conversation captures the alternating turns between the user and the agent, along with the evolving context of item mentions.

The ultimate goal of this work is to enable *personalized content generation* for the **target item** recommended by a chatbot in a multi-turn dialogue. Specifically, after interacting with a user across multiple conversational turns, the chatbot recommends a target item $i^*$. Based on the entire conversation history, the system aims to generate personalized content associated with $i^*$ that aligns with the user's preferences expressed during the conversation.

While the definition of "content" can be broad, in this paper, we focus on generating *visual content*, such as posters, thumbnails, or cover images. The generated visual should not only represent the semantics of the item $i^*$, but also incorporate stylistic or thematic elements that resonate with the user's individual taste. Formally, the task can be defined as: $v_{i^*}^* = \arg\max_{v \in \mathcal{V}_{i^*}} \text{Score}(v \mid C)$.

Ideally, the target $v_{i^*}^*$ should receive the highest scores across all metrics, such as similarity to the conversation history or the user's preferences. For visual content, the score can refer to the **semantic similarity** between the generated image [29] and the original image, or between the image and items the user has previously liked [35].

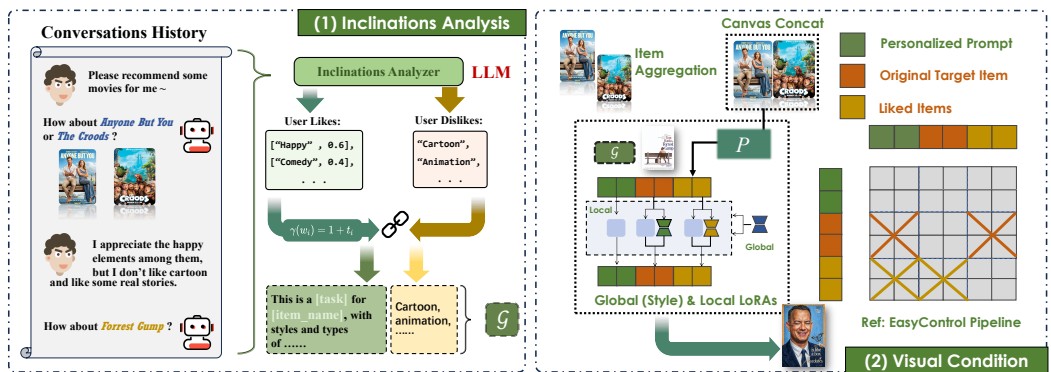

Figure 2: The PCG pipeline consists of two main components. On the left, the inclination analyzer summarizes the user's likes and dislikes. On the right, both global and local LoRAs control the style condition, the original target item condition, and the liked items condition. The personalized prompt has access to the other two tokens, while each of the two tokens is only aware of its own state.

# 4 Methods

In this section, we introduce the role of each component in generating the final output. First, the analyzer summarizes the user's conversation inclinations (*e.g.*, likes and dislikes), providing a deep understanding of past inclinations. Then, to better integrate previously preferred item content, we leverage DiTs' contextual understanding capabilities. Moreover, by using the sequence invariance of historical interactions under shuffled order, the model improves its ability to retrieve relevant interest-related content.

## 4.1 Inclinations Analyzer

During conversations, the user's inclinations primarily manifest in two ways: **(1) explicit statements**, such as "I want a xxx type of item" or "I don't like items with xxx style," and **(2) reactions** to the item candidates provided by the system, for instance, liking *Titanic* or *Flipped*, which indirectly suggests a preference for romantic items or items about love. These two types of preferences complement each other, and analyzing them in a unified manner is essential for accurately understanding the user's inclinations (both likes and dislikes).

Let the user's positive inclination (preference) word set be:

$$\mathcal{W} = \{w_1, w_2, \ldots, w_n\}. \tag{1}$$

Each word $w_i$ is associated with a non-negative preference score $t_i = s(w_i) \in \mathbb{R}_{\geq 0}$, which reflects the user's level of interest in that item. To normalize the scores into a valid probability distribution (as shown in Figure 2), we define the user's tendency distribution $\mathcal{T}$ as:

$$\mathcal{T} = \{(w_i, t_i) \mid w_i \in \mathcal{W}\}. \tag{2}$$

For example, in the movie poster scenario, the user's original preference scores could be:

$$\mathcal{T} = \{(\text{“Sci-fi”}, 0.375), (\text{“Warm”}, 0.25), (\text{“Mystery”}, 0.25), (\text{“Comedy”}, 0.125)\}.$$

The inclination analyzer converts users' explicit statements and implicit feedback into quantifiable preference weights through a *decay-aware weighting mechanism*. Given a tendency distribution $\mathcal{T} = \{(w_i, t_i)\}$, we compute the final recommendation weights as:

$$\gamma(w_i) = 1 + t_i. \tag{3}$$

This design provides two distinctive advantages over conventional normalization approaches: **Non-Zero Preservation**: By establishing a baseline weight of 1.0 for all attributes, our method prevents complete elimination of low-probability preferences. Formally, we have $\gamma(w_i) \geq 1.0, \quad \forall w_i \in \mathcal{W}$.

Therefore, even marginally preferred attributes (*e.g.*, $t_i < 0.1$) retain influence in the generation process. And for **Relative Emphasis Modulation**: The additive weighting compresses extreme differences while maintaining ordinal relationships. For any two attributes $(w_j, w_k)$:

$$\frac{\gamma(w_j)}{\gamma(w_k)} = \frac{1 + t_j}{1 + t_k} < \frac{t_j}{t_k}, \quad \text{when } t_j > t_k. \tag{4}$$

This property mitigates over-specialization in recommendations, especially in the process of generation process. After obtaining each $w_i$ and its corresponding $\gamma(w_i)$, we aim to ensure the generation model effectively captures this relationship. Studies show that DiT-based models can leverage (word, weight) pairs for prompt weight adjustment [19]. Hence, we design a **positive prompt** template, denoted as $\mathcal{G}_{\text{pos}}$, which can be formulated as:

$$\mathcal{G}_{\text{pos}} = \text{This is a [task] for [item\_name], with styles and types of } (w_1 : \gamma(w_1)), \ldots,$$

where $w_i \in \mathcal{W}$. In practice, we choose the top-$K$ highest weight words to construct the personalized prompt. For the **negative prompt** $\mathcal{G}_{\text{neg}}$, it mainly includes the types disliked by the user. This is used to reduce the appearance of elements that may cause discomfort. Since these cases are relatively rare, we directly use the disliked types to form the prompt without weighting the negative words.

This prompt acts as a textual supervision signal, guiding the generation process during the distribution mapping phase. $\mathcal{G} = \{\mathcal{G}_{\text{pos}}, \mathcal{G}_{\text{neg}}\}$ forms the overall **personalized prompt**.

However, relying solely on textual supervisory signals introduces several issues, such as the model's inability to capture the inherent semantic information of the original target item and integrate elements from previously liked item content. We additionally incorporate the content and textual information of both the target item and historically liked items for generation condition.

## 4.2 Visual Content Condition

To enable the model to condition on both the original item and historical liked items, it is essential to incorporate historical information into the generation process. Specifically, the personalized prompt should guide the generation jointly with the visual content, while maintaining as much independence as possible between the original item and historical items. This is because the generation process introduces personalized elements based on the original item. A straightforward approach is to use Group Diffusion Transformers [9]. However, such methods have two major drawbacks: they are heavily influenced by the VAE and fail to make historical visual elements visible to each other during the condition generation process.

Inspired by [10], we propose a method to allow textual and visual modeling to jointly guide the generation process without modifying any model architecture. Specifically, historical visual content is concatenated before encoding and then fed into the encoder as a single image. This approach is motivated by [10], which demonstrated that DiT can effectively handle concatenated images and merged prompts, capturing contextual relationships within them.

### 4.2.1 Multi-Item Aggregation

Given $N$ item images $\{I_i\}_{i=1}^{N}$ a user liked in the conversation, we spatially concatenate them into a unified canvas $I_{\text{cat}}$:

$$I_{\text{cat}} = \mathcal{C}(I_1, I_2, ..., I_N) \in \mathbb{R}^{H \times (N \cdot W) \times C}, \tag{5}$$

where $\mathcal{C}(\cdot)$ denotes row-wise concatenation operator. Similarly, we could also define a column-wise operator. In general, for vertically oriented visual items, we can stack them row by row for concatenation, while for horizontally oriented items, we concatenate them along the column dimension. This approach ensures that the aspect ratio of the resulting image does not vary significantly, enabling the text to better guide the image generation process. It is important to note that during the generation process, the fused information should not depend on the order of concatenation. Therefore, the training process should be designed to minimize sensitivity to the order of concatenation.

### 4.2.2 Sequence Invariance

To make the model more robust to the order of input features while keeping their meaning unchanged, we propose a mechanism called **Sequence Invariance**. This method ensures that the model's output does not depend on the order in which input items are concatenated. We implement this by randomly changing the order of concatenated input images during training.

We define the concatenated image $I_{\text{cat}}$ as a block matrix as in Equation 5 with $I_1$ to $I_N$. Each block $I_i \in \mathbb{R}^{H \times W \times C}$ is an individual input image, and $I_{\text{cat}} \in \mathbb{R}^{H \times (N \cdot W) \times C}$ represents all $N$ images placed side by side. To apply Sequence Invariance, we use a permutation matrix $P \in \{0,1\}^{N \times N}$ to reorder the blocks in $I_{\text{cat}}$:

$$I_{\text{cat, shuffled}} = P \cdot I_{\text{cat}}, \tag{6}$$

where $P$ satisfies:

$$P_{ij} = \begin{cases} 1 & \text{if block } j \text{ moves to position } i, \\ 0 & \text{otherwise,} \end{cases} \qquad P^\top P = PP^\top = I_N. \tag{7}$$

Here, $P$ is an orthogonal matrix, and $I_N$ is the $N \times N$ identity matrix. This operation permutes the positions of $I_1, I_2, \ldots, I_N$ without changing their content.

During training, a new permutation matrix $P$ is randomly sampled for each batch. $P$ is chosen uniformly from the set: $\mathcal{P}_N = \{P \in \{0,1\}^{N \times N} \mid P^\top P = I_N\}$. This ensures all possible input orders are equally used in training and helps the model learn order-independent features.

We construct $P$ from a random permutation $\pi \in S_N$, the set of all orderings of $N$ elements:

$$P_{ij} = \begin{cases} 1 & \text{if } \pi(j) = i, \\ 0 & \text{otherwise.} \end{cases} \tag{8}$$

At inference time, we use the identity matrix $P = I_N$ to keep the original input order:

$$I_{\text{cat, shuffled}} = P \cdot I_{\text{cat}} = I_N \cdot I_{\text{cat}} = I_{\text{cat}}. \tag{9}$$

In Appendix C.2, we provide an in-depth analysis of the Sequence Invariance.

### 4.2.3 Global and Local LoRA

Based on concatenated images, how these images guide the generation process is a key challenge. In typical text-to-image generation tasks, textual prompts alone often serve as the condition. While this enables diverse outputs, it may also result in large deviations from the original visual content. A common strategy in the community is to first train a task-specific LoRA, which, through fine-tuning, embeds useful prior knowledge into the model for better alignment with the task.

In our case, we aim to condition the generation not only on the original image but also on historical visual information. This motivates the introduction of two types of LoRA modules: a **global LoRA** to control the overall style and semantics, and a **local LoRA** that selectively influences the content branch to better fuse visual conditions. The global LoRA is obtained based on the target task, though integrating visual features into the local LoRA is non-trivial.

Thanks to the powerful plugin system of `EasyControl`, we can fully leverage this dual-conditioning strategy during generation. As shown in Figure 2, built on FLUX, `EasyControl` provides additional LoRA weights for branches such as subject and background, enabling the model to incorporate user-specific visual history. Meanwhile, the global LoRA is applied across noise, personalized prompts, and visual condition branches to ensure consistency with the task. By jointly using a task-specific global LoRA and a visually-oriented local LoRA, our framework effectively balances personalized preferences with the faithful rendering of target item content during generation. Specifically, we adopt the same architecture as `EasyControl`, where the tokens from the personalized prompt attend to both the subject and background through local LoRA. In contrast, the subject and background tokens only interact with their own content. This design is important for our task because it ensures that control is consistently guided by the prompt. It also keeps the blending of subject and background under textual supervision, preventing mutual interference.

**Global LoRA**  In this stage, the base model parameters $\theta$ are frozen, and only the global(style) LoRA parameters $\phi_{\text{style}}$ should be trained. Specifically, no visual or dialogue conditioning is used. Our model is based on FLUX.1-dev [2], which uses a flow-matching loss. Therefore, the same loss function is adopted as:

$$\mathcal{L}_{\text{style}} = \mathbb{E}_{t,\, x_0 \sim \mathcal{N}(0,I),\, x_1} \left\| u_{\theta+\phi_{\text{style}}}(x_t, t) - (x_1 - x_0) \right\|_2^2, \tag{10}$$

where $x_0$ denotes Gaussian noise, $x_1$ denotes the original data sample, and $x_t$ is an intermediate state obtained by interpolating between them, typically defined as $x_t = (1-t)x_0 + tx_1$. The style LoRA parameters $\phi_{\text{style}}$ could be optimized using the gradient $\nabla_{\phi_{\text{style}}} \mathcal{L}_{\text{style}}$.

**Training Local LoRA with Obtained Global LoRA**  After the style LoRA $\phi_{\text{style}}$ is obtained and fixed, we introduce two local LoRA modules used for conversation context, $\phi_{\text{subj}}$ (weights are set from `EasyControl`, used in inference) and $\phi_{\text{bg}}$, to enable personalized visual conditioning. $F_\Theta(C)$, encoding conversation history information including user inclinations and concatenated images, guides the generation process. The generation is conditioned on these cues and can be optimized via:

$$\mathcal{L}_{\text{local}} = \mathbb{E}_{t,\, x_0 \sim \mathcal{N}(0,I),\, x_1,\, F_\Theta(C)} \left\| u_{\theta+\phi_{\text{bg}}}(x_t, t, F_\Theta(C)) - (x_1 - x_0) \right\|_2^2, \tag{11}$$

where gradients are applied to $\phi_{\text{bg}}$ (*i.e.*, $\nabla_{\phi_{\text{bg}}} \mathcal{L}_{\text{local}} \neq 0$), while $\phi_{\text{style}}$ remains frozen. In our setting, the LoRA weights are also initialized using `EasyControl` and can be further fine-tuned with the loss defined above.

**Inference**  During the inference process, it is necessary to balance the influence strength of the subject and background elements on the generated content. Let the weights of these two local LoRAs be represented by $\lambda_1$ and $\lambda_2$, respectively. Generally, $\lambda_1$ is set to be properly larger than $\lambda_2$ since the background serves as a supplementary role. We generally set $\lambda_1 = 1$ and $\lambda_2 = 0.85$ by default, while careful tuning in the possible vicinity of these values may further improve performance.

## 5  Experiment

In this section, we introduce the base model, comparison models, and details of the datasets used in our experiments. Although many conversational systems exist in the field, high-quality conversation data with available visual content **remains scarce**. Following previous works of conversational recommender systems [23, 40], we conduct experiments on two conversational recommendation datasets set in movie scenarios. The hyperparameters we adopted are shown in Appendix G.

### 5.1  Dataset

| Dataset | Dialogs | Utterances | Avg. Turns |
|---|---|---|---|
| E-ReDial | 756 | 12,003 | 15.9 |
| Inspired | 1,001 | 35,811 | 10.73 |

Table 1: Data overview of conversational datasets in the movie scenario.

The two datasets are classic benchmarks for movie conversational recommender systems, containing many high-quality interactions with the systems.

For both datasets, the original data was randomly split into training, validation, and test sets with a ratio of 8:1:1. The generated results were evaluated on the test set.

### 5.2  Evaluation Metrics

Moreover, we use CLIP to measure similarity between images and texts (CLIP-T), as well as between images (CLIP-I), in order to evaluate the semantic and visual similarity between the generated and original images. The Deep Inception Score (DIS) Score is also defined to assess the cosine similarity of the last layer of two images after passing through the Inception model.

In our evaluation, different image similarity metrics are computed using various image resolutions. Specifically, Inception-based metrics such as FID and DIS are processed at the resolution, while CLIP-based metrics (CLIP score and Image CLIP score) and LPIPS are evaluated at $224 \times 224$. For MS-SSIM, we also maintain the resolution for image pairs to capture fine-grained structural similarities. Moreover, for LPIPS calculation, we use AlexNet as the base model.

---

[2] `https://huggingface.co/black-forest-labs/FLUX.1-dev`

Table 2: The objective evaluation results of E-ReDial.

| Types | Models | Historical Liked | | Target | | | | | FID↓ |
|---|---|---|---|---|---|---|---|---|---|
| | | MS-SSIM↑ | CLIP-T↑ | DIS↑ | CLIP-T↑ | CLIP-I↑ | MS-SSIM↑ | LPIPS↓ | |
| Text-to-Image | SD3 | 0.1285 | 26.52 | 84.97 | 28.82 | 74.14 | 0.1286 | 0.6831 | 21.32 |
| | FLUX.1-dev | 0.1741 | 25.02 | 82.56 | 27.49 | 68.14 | 0.1875 | 0.6692 | 26.51 |
| | EasyControl-SUBJ | 0.1437 | 25.97 | 87.94 | 30.72 | 77.92 | 0.2166 | 0.5561 | 20.20 |
| Image-to-Image | EasyControl-BG | 0.1514 | 24.62 | 80.20 | 26.96 | 64.72 | 0.1765 | 0.6742 | 25.01 |
| | PCG (Ours) | 0.1589 | 26.52 | 85.76 | 30.52 | 73.94 | 0.1938 | 0.6238 | 21.88 |

Table 3: The objective evaluation results of Inspired.

| Types | Models | Historical Liked | | Target | | | | | FID↓ |
|---|---|---|---|---|---|---|---|---|---|
| | | MS-SSIM↑ | CLIP-T↑ | DIS↑ | CLIP-T↑ | CLIP-I↑ | MS-SSIM↑ | LPIPS↓ | |
| Text-to-Image | SD3 | 0.1145 | 26.12 | 84.24 | 27.81 | 69.31 | 0.1236 | 0.6810 | 20.72 |
| | FLUX.1-dev | 0.1819 | 25.36 | 83.25 | 27.17 | 68.32 | 0.1767 | 0.6673 | 25.66 |
| | EasyControl-SUBJ | 0.1521 | 25.87 | 88.52 | 30.70 | 79.08 | 0.2201 | 0.5371 | 19.74 |
| Image-to-Image | EasyControl-BG | 0.1906 | 24.89 | 81.09 | 26.38 | 61.62 | 0.1746 | 0.6947 | 25.41 |
| | PCG (Ours) | 0.1764 | 26.23 | 85.55 | 29.56 | 72.20 | 0.1872 | 0.6291 | 22.41 |

## 5.3 Main Experiment (Objective Evaluation)

The main experiment primarily involves the analysis of objective metrics related to the content. We focus on two key aspects: the integration of historical liked items into the generated content and the fidelity to the target item intended for interaction. Specifically, for `EasyControl`, we evaluate the generative performance in two separate settings: **-SUBJ**, where only information from the target item is used for conditioning, and **-BG**, where only background information is used as the conditioning input. Both settings are based on the `EasyControl` framework. The experiment on E-ReDial and Inspired can be found in Table 2 and 3.

As shown in Table 2, the conventional text-to-image methods struggle to maintain high fidelity to the target item. Specifically, the similarity between the generated movie posters and the captions of the target item's posters, measured by CLIP-T, is around 27–29, generally lower than that achieved by image-to-image models. This highlights the importance of effectively conditioning on the target item to ensure model fidelity. On the other hand, text-to-image models perform relatively well in integrating historical interests. For example, the FLUX.1-dev model achieves high SSIM similarity scores, and SD3 outperforms in the CLIP-T metric when compared to historical liked items. For EasyControl-BG, it can be observed that solely using concatenated historical items does not improve historical relevance compared to FLUX.1-dev. This indicates that a single LoRA for background signals cannot generate a coherent guiding signal with the prompt, meaning it fails to maintain fidelity or integrate historical interests.

On E-ReDial, when using the target item directly as the image condition along with user historical items as text prompts, the historical information can be partially integrated. This approach achieves an MS-SSIM of 0.1437 against historical images, showing noticeable improvement compared to SD3. However, the integration is still not thorough enough, as evidenced by the two historical interest metrics where EasyControl-SUBJ underperforms compared to our proposed PCG framework. This further validates the effectiveness of our method. On Inspired in Table 3, PCG excels at preserving historical user preferences. While the baseline FLUX.1-dev can partially reflect user history, its semantic fidelity (measured by CLIP-T) is significantly lower than PCG's. EasyControl-SUBJ shows strong performance in capturing details of the target item, but this comes at the cost of weaker integration of information from previously liked items. These results highlight PCG's ability to strike a strong balance between fidelity and personalization.

Overall, text-to-image methods exhibit promising results in capturing historical interests, likely because explicitly incorporating user history into prompts enhances generative capability. In contrast, image-to-image methods demonstrate superior fidelity, possibly due to stronger perceptual alignment with image signals. Achieving optimal performance in both aspects simultaneously is challenging, as there is an inherent trade-off between them. Nevertheless, PCG effectively integrates historical information while maintaining high fidelity, proving its overall superiority.

Table 4: GPT evaluation score (subset).

| Model | V | I | T | D |
|---|---|---|---|---|
| FLUX.1-dev | 3.00 | 2.00 | 2.25 | 2.75 |
| SD-3 | 4.50 | 4.50 | 4.75 | 4.50 |
| EC-SUBJ | 2.00 | 1.25 | 1.50 | 2.25 |
| EC-BG | 3.25 | 3.25 | 3.00 | 3.75 |
| PCG | 5.00 | 5.00 | 4.75 | 5.00 |

Table 5: GPT evaluation score (full test set).

| Model | V | I | T | D |
|---|---|---|---|---|
| FLUX.1-dev | 3.94 | 3.23 | 3.26 | 3.97 |
| SD-3 | 3.48 | 3.13 | 3.11 | 3.40 |
| EC-SUBJ | 4.14 | 3.82 | 3.86 | 4.03 |
| EC-BG | 3.81 | 3.61 | 3.31 | 3.95 |
| PCG | 4.34 | 4.01 | 4.17 | 4.38 |

## 5.4 GPT Evaluation

Due to the subjective nature of personalized generation, we use GPT-4o to evaluate the generated results, aiming to make the assessment more objective and targeted. When GPT-based models are unavailable, the Gemini series models serve as good alternatives. In this part of the experiment, we first randomly select a subset of generated dialogues from the test set and compare them with other generated results. The relevant prompts are provided in Appendix I. According to the scores from GPT on E-ReDial subset (shown in Table 4), our proposed PCG performs well across all evaluation aspects, indicating a good overall balance between fidelity and personalization. This gap is substantial, demonstrating the exceptional performance of PCG in specific contexts and scenarios.

On the full E-ReDial test set, PCG shows a smaller score gap compared to other base models. However, PCG still achieves the best performance across all four dimensions. Apart from PCG, EC-SUBJ closely follows in each dimension, showing strong consistency with the objective metrics evaluation. For example, its performance in objective metrics is outstanding, and it also scores highly in subjective evaluation, particularly in coherence and details. EC-BG performs well in the Integration dimension, indicating that background as a condition is effective. However, due to a lack of good synergy with other techniques, it slightly underperforms compared to the base FLUX.1-dev model in visual presentation metrics.

## 5.5 Hyperparameter Analysis

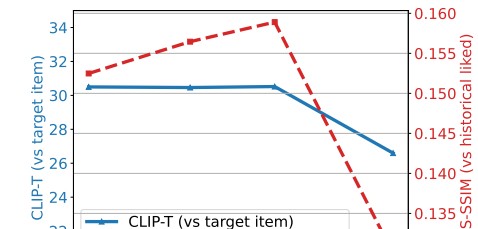

Figure 3: The impact of $\lambda_2$ on E-ReDial dataset.

We analyze the impact of $\lambda_1$ and $\lambda_2$ on the generated results. In our experiments, we fix $\lambda_1 = 1.0$ and vary $\lambda_2$, which controls the background strength. Overall, increasing $\lambda_2$ improves both fidelity and personalization. However, when $\lambda_2 > 1$, the performance drops significantly. Notably, the CLIP-T vs. target item score remains stable for $\lambda_2 < 1$, but shows a clear decline at $\lambda_2 = 1.2$. This is likely because the background becomes too dominant, leading to visual clutter, or what we refer to as chaotic "patterned images." Thus, selecting an appropriate $\lambda_2$ is crucial to balance fidelity and personalization for different subjects. Further hyperparameter analysis of it, and the impact of training steps on LoRA are provided in Appendix F.1.

## 5.6 Visual Comparison

In Figure 4, we present the generation results of five models based on four conversations. Two of the original conversations are shown in Appendix J. It can be observed that PCG achieves a good balance between fidelity to the original image and personalization.

For example, in the second row, with the case of *Superbad*, the user's favorite movies include typical teen comedies such as *Meet the Parents* and *The Hangover*. PCG preserves the high-school vibe and playful tone of *Superbad*, while enhancing the character positions and background colors in the

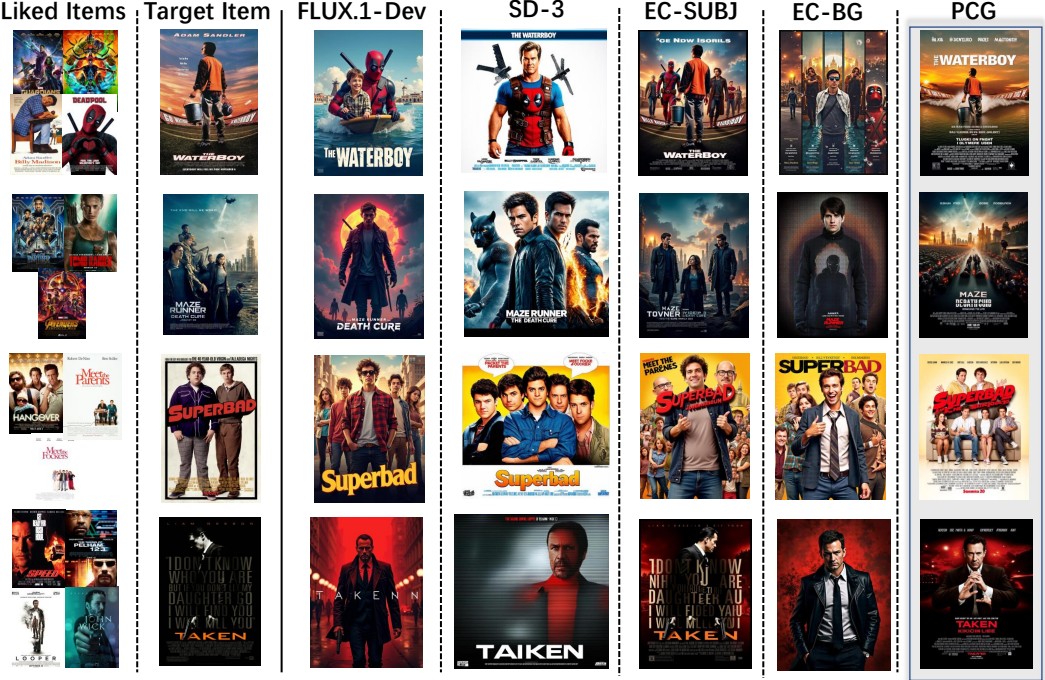

Figure 4: Visual Comparison of PCG with other models. Here EC represents `EasyControl`.

poster, making it more vibrant and youthful. In contrast, the outputs of other models show significant style changes or simply replace the background without maintaining the original atmosphere. In the fourth row, the target item is the classic *Taken* poster, which features a strong cold tone, tense atmosphere, and the iconic composition of the protagonist standing alone. PCG retains the dark tone and tension of the original *Taken* poster while integrating the frontal portrait style of historical items, effectively capturing elements of the user's inclinations.

Overall, PCG generates results that naturally combine historical interests while maintaining high fidelity to the target item. In comparison, EC-SUBJ performs similarly, but it is slightly less effective in capturing the overall style and transferring the atmosphere.

## 6 Conclusion

To address the lack of visual diversity in conversational systems, we propose **PCG**, a novel personalized generation framework that integrates user preferences while maintaining fidelity to the original item. Specifically, PCG leverages the strong language understanding capabilities of large models to efficiently infer user likes and dislikes, which are then used to construct personalized prompts guiding the generation process. In addition, to better incorporate visual elements that users may prefer, we introduce a visual content conditioning method. This method uses both global and local LoRAs to align the generation with the target item and historically liked items. We evaluate PCG on standard benchmarks, focusing on both fidelity to the target item and personalization based on user history. Our results demonstrate that PCG show great generation ability, which may offer a promising new direction for improving user experience in conversational systems.

## Acknowledgment

This research was partially supported by the National Natural Science Foundation of China (Grants No. 62406303, 62525606), the Anhui Province Science and Technology Innovation Project (202423k09020010), and the USTC Bihe Youth Program for Interdisciplinary Innovation (BH-202518).

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

# Appendix

## A   Limitations

Although our method demonstrates its effectiveness through subjective and objective evaluations, hyperparameter analysis, ablation experiments, and result visualizations, it has some limitations. The approach is highly dependent on the quality of the original images (fortunately, the images used in this study are of high resolution), the ability of the base LLM for interest analysis, and its relatively slow generation speed, which can be a challenge for real-time conversations.

We hope that the ideas presented in this paper will offer new insights for personalized content generation in conversational systems, pave the way for further developments, and inspire more excellent work. In our future work, we will also focus on refining the generation of human-related details and addressing the current issue where imperfect content fidelity may cause generated outputs to stray somewhat from the original intent.

## B   Broader Impact

The proposed visual content generation framework, PCG, demonstrates a significant advancement in personalized content synthesis by leveraging user preferences to create visually compelling and contextually aligned content. By integrating liked items with the target movie, PCG enhances visual consistency and thematic relevance, bridging the gap between user-specific taste and cinematic representation. This approach not only amplifies user engagement but also introduces novel ways for users to interact with media content, offering a more immersive and personalized experience.

The main advantage of PCG lies in its ability to accurately capture thematic elements from both the target movie and the user's preferred visual styles. Unlike traditional generation methods that merely focus on stylistic transfer, PCG effectively balances semantic alignment with visual aesthetics, leading to outputs that are not only artistically pleasing but also contextually meaningful. This capability opens up new opportunities in content marketing, fan engagement, and interactive media, allowing for dynamically generated visuals that resonate with individual tastes.

However, the method also introduces potential ethical considerations. Personalized content generation, if not regulated properly, may inadvertently promote biased representations based on user history, reinforcing stereotypes or unbalanced views. Furthermore, the use of copyrighted materials for personalized synthesis raises questions about intellectual property rights and fair use, particularly when generated content resembles original artworks. Addressing these challenges requires clear guidelines on data privacy, responsible use of AI in media creation, and proper attribution for derivative works to ensure ethical and fair application.

## C   Theoratical Support for PCG

### C.1   Flow Matching Loss Formulation

#### C.1.1   Loss Analysis on Global LoRA

We provide the flow matching loss on Global LoRA as:

$$\mathcal{L}_{\text{style}} = \mathbb{E}_{t,\, x_0 \sim \mathcal{N}(0,I),\, x_1} \left\| u_{\theta + \phi_{\text{style}}}(x_t, t) - (x_1 - x_0) \right\|_2^2. \tag{12}$$

This loss function trains the LoRA parameters $\phi_{\text{style}}$ to predict the velocity field that transforms noise samples $x_0 \sim \mathcal{N}(0, I)$ into data samples $x_1 \sim p_{\text{data}}$. The intermediate state $x_t$ is defined through linear interpolation:

$$x_t = (1 - t)x_0 + tx_1. \tag{13}$$

The gradient with respect to the style LoRA parameters $\phi_{\text{style}}$ is computed as:

$$\nabla_{\phi_{\text{style}}} \mathcal{L}_{\text{style}} = \mathbb{E}_{t, x_0 \sim \mathcal{N}(0,I), x_1} \left[ 2(u_{\theta + \phi_{\text{style}}}(x_t, t) - (x_1 - x_0)) \cdot \nabla_{\phi_{\text{style}}} u_{\theta + \phi_{\text{style}}}(x_t, t) \right]. \tag{14}$$

This gradient drives the parameter updates through gradient descent:

$$\frac{d\phi_{\text{style}}}{d\tau} = -\nabla_{\phi_{\text{style}}}\mathcal{L}_{\text{style}}. \tag{15}$$

Under the LoRA parameterization $\phi_{\text{style}} = BA$, where $B \in \mathbb{R}^{d \times r}$ and $A \in \mathbb{R}^{r \times d}$, the gradients decompose as:

$$\nabla_A \mathcal{L}_{\text{style}} = B^T \nabla_{\phi_{\text{style}}}\mathcal{L}_{\text{style}}, \tag{16}$$

$$\nabla_B \mathcal{L}_{\text{style}} = \nabla_{\phi_{\text{style}}}\mathcal{L}_{\text{style}} A^T. \tag{17}$$

This low-rank parameterization constrains updates to a compact subspace, enabling efficient and modular style adaptation while preserving the base model's core capabilities.

### C.1.2 Application to Local LoRAs

The gradient flow for Local LoRA analysis is identical to that of Style LoRA and will not be repeated here.

### C.2 Sequence Invariance as Flow Matching Regularization

In our work, *how sequence invariance affects the training or inference process* remains an open question. In fact, the sequence invariance strategy can be interpreted as a regularization method in the flow matching framework.

**Proposition 1** (Sequence Invariance Regularization). *Let $\pi \in S_N$ be any permutation of the input source sequence. Define the permuted source distribution as $q_0^\pi$. The ideal goal is:*

$$\mathcal{L}_{FM}(q_0^\pi, q_1) = \mathcal{L}_{FM}(q_0, q_1), \quad \forall \pi \in S_N, \tag{18}$$

*where $\mathcal{L}_{FM}$ is the flow matching loss.*

*Introducing random permutation matrices $P$ during training is equivalent to minimizing the expected flow matching cost:*

$$\mathbb{E}_{\pi \sim Uniform(S_N)}[\mathcal{L}_{FM}(q_0^\pi, q_1)]. \tag{19}$$

*This encourages the model to learn representations that are invariant to input orderings.*

### C.2.1 Group Theory Perspective on Sequence Invariance

From a group theory standpoint, sequence invariance corresponds to invariance under the symmetric group $S_N$.

A function $f$ is sequence-invariant if and only if:

$$f(P \cdot x) = f(x), \quad \forall P \in \mathcal{P}_N. \tag{20}$$

This can be enforced using the Reynolds operator:

$$\hat{f}(x) = \frac{1}{N!} \sum_{\pi \in S_N} f(P_\pi \cdot x). \tag{21}$$

This suggests that random permutations during training project the model toward the invariant function space.

### C.3 The Choice of $\lambda_1$ and $\lambda_2$.

Let $\lambda_1 = 1$ and $\lambda_2 = 0.85$ be the default fusion weights used during inference. Here we prove why we could set the $\lambda_1$ for Global LoRA and change $\lambda_2$. These settings can be justified using Bayesian perspective and Information Bottleneck view.

### C.3.1 Bayesian Optimal Weight Allocation for Multimodal Information Fusion

To explain why we set $\lambda_1 > \lambda_2$, we analyze the theoretical rationale behind this choice from a Bayesian perspective. Given subject data $D_{\text{subj}}$ and background data $D_{\text{bg}}$, the target is to estimate the posterior $p(x|D_{\text{subj}}, D_{\text{bg}})$. By Bayes' rule:

$$p(x|D_{\text{subj}}, D_{\text{bg}}) \propto p(D_{\text{subj}}|x)p(D_{\text{bg}}|x)p(x). \tag{22}$$

Taking logarithms and it could be obtained:

$$\log p(x|D_{\text{subj}}, D_{\text{bg}}) \propto \log p(x) + \log p(D_{\text{subj}}|x) + \log p(D_{\text{bg}}|x). \tag{23}$$

Assuming these are parameterized by neural networks, the optimal fusion weights are inversely proportional to the conditional entropy:

$$\lambda_1 \propto \frac{1}{\mathcal{H}(D_{\text{subj}}|x)}, \quad \lambda_2 \propto \frac{1}{\mathcal{H}(D_{\text{bg}}|x)}. \tag{24}$$

Since subject data usually provides more certain information, this supports the setting $\lambda_1 = 1 > \lambda_2$.

### C.3.2 Information Bottleneck View of Fusion

To explain why we fix $\lambda_1 = 1$ and vary $\lambda_2$, we provide a proof from the Information Bottleneck perspective, showing that the ratio between the two (rather than their specific values) has the greatest impact on the fusion data. From the information bottleneck principle, optimal fusion maximizes relevant information while minimizing redundancy:

$$\max_{\phi} I(Z; Y) - \beta I(Z; X), \tag{25}$$

where $X$ represent the input data, $Y$ represent the target output, and $Z$ represent the learned representation. We define $\beta$ as an entropy-weighted trade-off parameter. Then the ratio of fusion weights satisfies:

$$\frac{\lambda_1}{\lambda_2} \approx \frac{I(Z_{\text{subj}}; Y)}{I(Z_{\text{bg}}; Y)} \cdot \frac{1 - \beta_{\text{bg}}}{1 - \beta_{\text{subj}}}. \tag{26}$$

## D  Pseudo Code

### D.1  Training Pseudo Code

---
**Algorithm 1** Training Process with Sequence Invariance
---

1: **Input:** $N$ item images $\{I_i\}_{i=1}^N$, historical content
2: **Step 1:** Spatially concatenate images to form $I_{\text{cat}} \in \mathbb{R}^{H \times (N \cdot W) \times C}$
3: **Step 2:** Generate permutation matrix $P$ via:
4:     Sample $\pi \sim S_N$ & construct $P$ where $P_{ij} = \mathbb{I}[\pi(j) = i]$
5:     Apply shuffling: $I_{\text{cat, shuffled}} = P \cdot I_{\text{cat}}$
6: **Step 3:** Obtain Global LoRA $\phi_{\text{style}}$ [3]
7: **Step 4:** Fine-tune Local LoRA $\phi_{\text{bg}}$:
8:     Freeze $\phi_{\text{style}}$
9:     Optimize $\phi_{\text{bg}}$ using local loss $\mathcal{L}_{\text{local}}$ with conversational (text/image) context
10: **Step 5:** Update parameters of LoRA using gradient descent

---

[3]In *movie* conversational system scenarios, to facilitate reproducibility and reduce training resource requirements, a global LoRA can be adopted from `https://civitai.com/models/786013/flux-lora-movie-poster-modern`.

## D.2 Inference Pseudo Code

The inference phase involves several modules, including user inclinations analysis, integration of historical items, as well as flow matching transformation and decoding processes. The following pseudo code illustrates the main parts of these processes in detail.

---

**Algorithm 2** Inference Process with Global and Local LoRA

---

1: **Input:** Dialogue history $C$, reference images $\{I_{\text{ref}}\}$ including $\{I_{\text{cat}}\}$
2: **Step 1:** Load pretrained components:
3:      Base model parameters $\theta$ (FLUX.1-dev)
4:      Global LoRA $\phi_{\text{style}}$ (frozen, pretrained)
5:      Local LoRAs: $\phi_{\text{subj}}$, $\phi_{\text{bg}}$ (fine-tuned)
6: **Step 2:** Set LoRA weights:
7:      $\lambda_1 = 1.0$ // Subject LoRA weight
8:      $\lambda_2 = 0.85$ // Default background LoRA weight (supplementary signals)
9: **Step 3:** Analyze inclinations:
10:      $\mathcal{G} = \text{InclinationsAnalyzer}(C)$ // Extract text inclinations
11: **Step 4:** Initialize noise and timestep:
12:      Sample $x_0 \sim \mathcal{N}(0, I)$ // Start from Gaussian noise at $t = 0$
13:      Set timestep schedule $\{t_i\}_{i=0}^{T}$ with $t_0 = 0, t_T = 1, t_i \in \mathbb{R}$
14: **Step 5:** Flow Matching inference loop:
15: **for** $i = 0$ **to** $T - 1$ **do**
16:      Combine LoRA parameters:
17:          $\phi_{\text{combined}} = \phi_{\text{style}} \cup \lambda_1 \cdot \phi_{\text{subj}} \cup \lambda_2 \cdot \phi_{\text{bg}}$
18:      Apply `EasyControl` framework:
19:          Inject personalized prompt $\mathcal{G}$ to both subject & background branches with $\{I_{\text{ref}}\}$
20:          Ensure subject/background tokens interact within own content
21:      Predict velocity field:
22:          $u_{t_i} = u_{\theta + \phi_{\text{combined}}}(x_{t_i}, t_i, F_\Theta(C))$ // $t_i \in \mathbb{R}$
23:      Update latent via Euler method:
24:          $\Delta t = t_{i+1} - t_i$
25:          $x_{t_{i+1}} = x_{t_i} + \Delta t \cdot u_{t_i}$
26: **end for**
27: **Step 6:** Decode final image:
28:      $I_{\text{generated}} = \text{Decoder}(x_1)$ // $x_1$ is the sample at $t = 1$
29: **Output:** Generated image $I_{\text{generated}}$

---

# E  Contextual Understanding Through Concatenation

We suggest that concatenation allows the DiTs model to capture the semantic information of previously liked items. Although [10] provides empirical evidence of its effectiveness, it lacks a deep analysis of the underlying mechanism. Here, we explain how DiTs integrates semantic context through concatenation by analyzing how the transformer's attention mechanism interprets background information.

## E.1  Context Modeling Ability of Transformer Attention Mechanism

DiT, which is based on the Transformer architecture, leverages the Self-Attention mechanism that naturally captures long-range dependencies. When input sequences, such as image features or text tokens, are concatenated, each token can attend to every other token in the sequence through attention weights. This process enables the establishment of global semantic relationships. Specifically, the model computes a Query vector for each token, which is compared to the Key vectors of all other tokens to calculate similarities. The resulting similarity scores are used to weight and aggregate the corresponding Value vectors. This allows the model to focus on key information within the sequence, such as object features or scene styles, and incorporate this information into the current task. Additionally, the multi-head attention mechanism decomposes the attention computation into

multiple subspaces, each focusing on different dimensions of context (*e.g.*, color, spatial arrangement). These multi-dimensional semantic insights are then integrated to generate coherent outputs.

## E.2 Input Concatenation Strategy Activating the Model's Contextual Representation Ability

DiT activates the model's contextual representation ability by concatenating image or text tokens, providing explicit contextual cues. Unlike traditional methods that concatenate only attention tokens, DiT directly concatenates multiple images into a single input. This approach preserves both spatial and semantic relationships between images in the feature space. It enables the model to understand the overall theme of the image set and maintain consistency in style and logic during generation. In the case of PCG's Personalized prompt design, the model extracts a global semantic framework through a text encoder and transforms it into contextual constraints for image generation. Furthermore, pretraining allows the model to link abstract concepts in text with image features. When multiple image prompts are provided, the model can use its pretraining knowledge to infer the semantic relationships between the images. As a result, pretraining DiT enables efficient multi-image understanding through *In-Context Learning*.

# F More Experiments

## F.1 Hyperparameter Analysis

In this section, we conduct an additional hyperparameter analysis.

### F.1.1 Impact of Local LoRA Training Steps on Generation Quality

Table 6: The influence of Local LoRA training steps on E-ReDial.

| Steps | Historical Liked | | Target | | | | | FID↓ |
|---|---|---|---|---|---|---|---|---|
| | MS-SSIM↑ | CLIP-T↑ | DIS↑ | CLIP-T↑ | CLIP-I↑ | MS-SSIM↑ | LPIPS↓ | |
| 100 | 0.1470 | 26.38 | 86.15 | 30.39 | 76.01 | 0.1768 | 0.6354 | 21.91 |
| 300 | 0.1568 | 26.64 | 85.94 | 30.37 | 74.76 | 0.1770 | 0.6448 | 22.89 |
| 500 | 0.1542 | 26.54 | 86.23 | 30.30 | 75.47 | 0.1773 | 0.6393 | 22.67 |
| 700 | 0.1511 | 26.68 | 86.03 | 30.48 | 74.71 | 0.1689 | 0.6445 | 22.86 |

The number of training steps for LoRA significantly affects the generation results. If too large, it may lead to overfitting, while if too small, it may fail to capture the fusion of background information, lacking relevant capabilities. Therefore, during training, we select the most suitable checkpoint based on the generation performance on the validation set, which is then used for generation on the test set.

From Table 6, it can be observed that as the number of training steps increases, the model's ability to fuse historical liked items initially increases and then decreases, but overall it shows an upward trend. Comparing the MS-SSIM and CLIP-T scores relative to historical items, we see an overall improvement in history fusion capability compared to training for 100 steps. However, there is a noticeable decline in subject fidelity, as shown by the decrease in DIS and CLIP-related metrics for the target item. This indicates that achieving a balance between personalization and fidelity requires selecting the most appropriate training steps based on the validation set for different tasks.

### F.1.2 Deeper Impact of Variations in $\lambda_1$ and $\lambda_2$

In Section 5.5 of the main text, we have analyzed the impact of different values of $\lambda_2$ on quantitative metrics, and here we present its effect on generation quality. As shown in Figure 6, the generated results vary significantly with changes in $\lambda_2$. When $\lambda_2$ is around 0.3, the generated image incorporates some information from the liked items, but it still primarily reflects elements related to the target item. As $\lambda_2$ increases to around 0.6–0.8, a balance is achieved between fidelity to the target and personalization. However, when $\lambda_2$ exceeds 1, the generated result is mainly a combination of liked items, as the strong guidance from background information limits the reflection of the original target item.

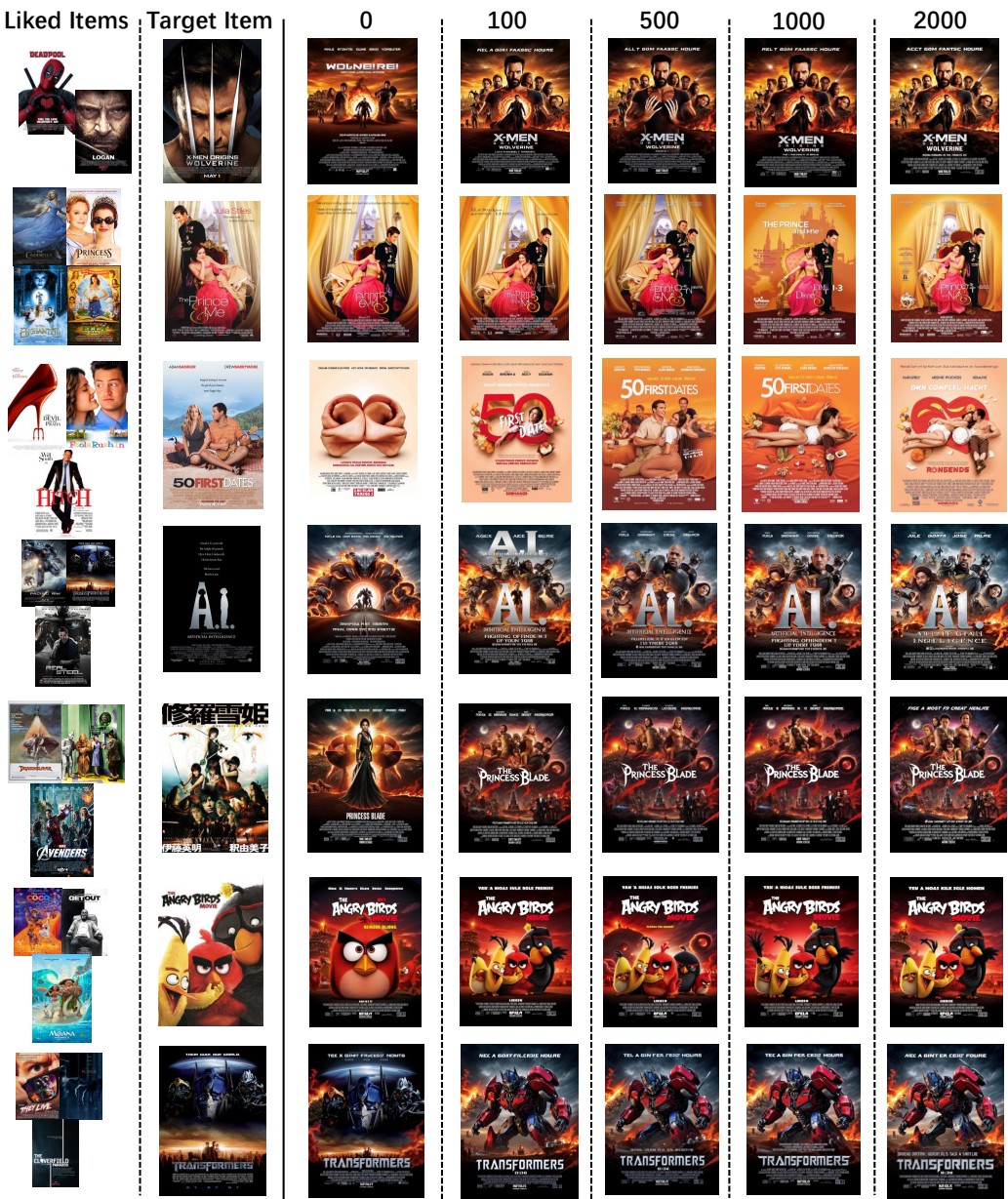

Figure 5: Generation results with different training steps for Local LoRA.

## F.2 Ablation Studies

In this chapter, we conduct various ablation studies on components and strategies, such as Sequence Invariance, to demonstrate the effectiveness of each part.

### F.2.1 The Impact of Sequence Invariance

In this experiment, we examine the impact of using (w/) and not using (w/o) Sequence Invariance (SI) on quantitative metrics, under the setting of training for 1000 steps with Local LoRA. As shown in Table 7, after applying SI, the generated images better integrate historical information, achieving higher scores on MS-SSIM and CLIP-T. Additionally, SI improves fidelity, as reflected in better

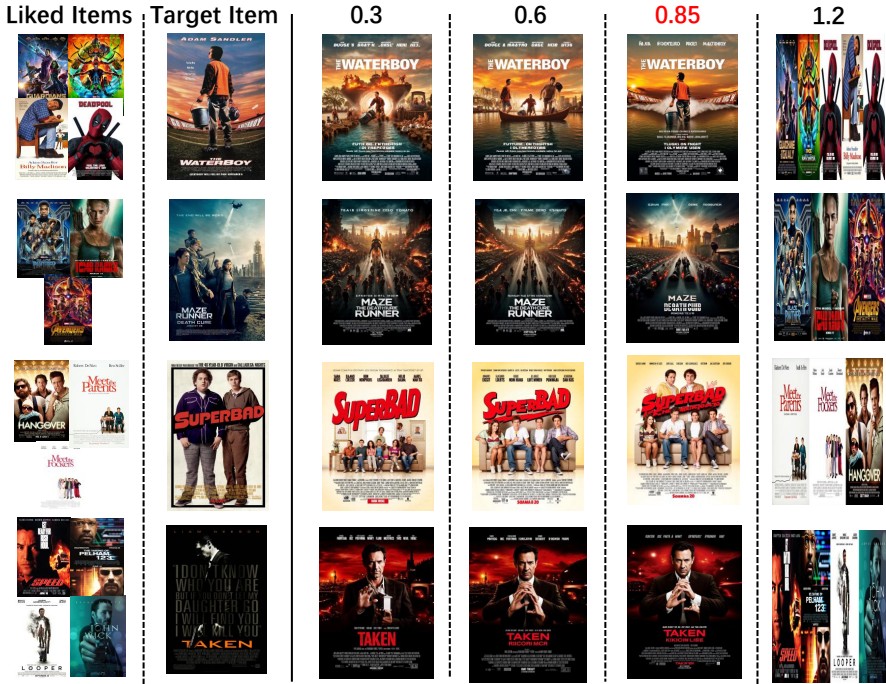

Figure 6: The generation results with different $\lambda_2$.

performance on metrics like SSIM. While these differences may not be immediately noticeable in the generated results, the metrics clearly demonstrate the effectiveness of SI.

Table 7: The influence of Sequence Invariance on E-ReDial.

| Method | Historical Liked | | Target | | | | | FID↓ |
|---|---|---|---|---|---|---|---|---|
| | MS-SSIM↑ | CLIP-T↑ | DIS↑ | CLIP-T↑ | CLIP-I↑ | MS-SSIM↑ | LPIPS↓ | |
| 1000 steps w/o SI | 0.1464 | 26.53 | 85.92 | 30.33 | 75.67 | 0.1670 | 0.6496 | 22.68 |
| 1000 steps w/ SI | 0.1488 | 26.69 | 85.92 | 30.34 | 75.52 | 0.1710 | 0.6487 | 22.49 |

### F.2.2 The Impact of Main Components

In this section, we focus on the impact of the following three main components on generation quality metrics: Global LoRA, Negative Inclinations and Score Reweighting. These components play a crucial role in both the fidelity and personalization of the generated results.

Table 8: The ablation study of E-ReDial.

| Method | Historical Liked | | Target | | | | | FID↓ |
|---|---|---|---|---|---|---|---|---|
| | MS-SSIM↑ | CLIP-T↑ | DIS↑ | CLIP-T↑ | CLIP-I↑ | MS-SSIM↑ | LPIPS↓ | |
| w/o Global LoRA | 0.1521 | 26.45 | 87.29 | 29.11 | 71.31 | 0.1890 | 0.5993 | 21.13 |
| w/o Negative Prompt | 0.1601 | 26.47 | 85.63 | 30.55 | 73.89 | 0.1945 | 0.6257 | 21.98 |
| w/o Score Reweight | 0.1594 | 26.46 | 85.72 | 30.39 | 73.39 | 0.1970 | 0.6250 | 21.92 |

From the ablation study in Table 8, we find that Global LoRA is the most important component, playing an essential role in both personalization and fidelity. This is because generating movie posters requires a strong community/pre-trained LoRA to provide relevant guidance; without it, the model struggles to meaningfully connect content with the poster. Next, the Score Reweighting mechanism significantly impacts the fidelity of the generated results. Without Score Reweighting, the generated outputs, compared to target items, show a decrease in quality, and the CLIP-T score relative to historical liked items also drops. This indicates that reweighting helps generate higher-quality content.

Lastly, the impact of Negative Prompt is minimal, likely because other components compensate for the negative influence on generation.

In terms of visualizing the generated results, we can clearly observe that Global LoRA has the most significant impact on the generation quality, while the effects of Negative Prompt and Score Reweighting are less apparent visually, despite the differences shown in the evaluation metrics.

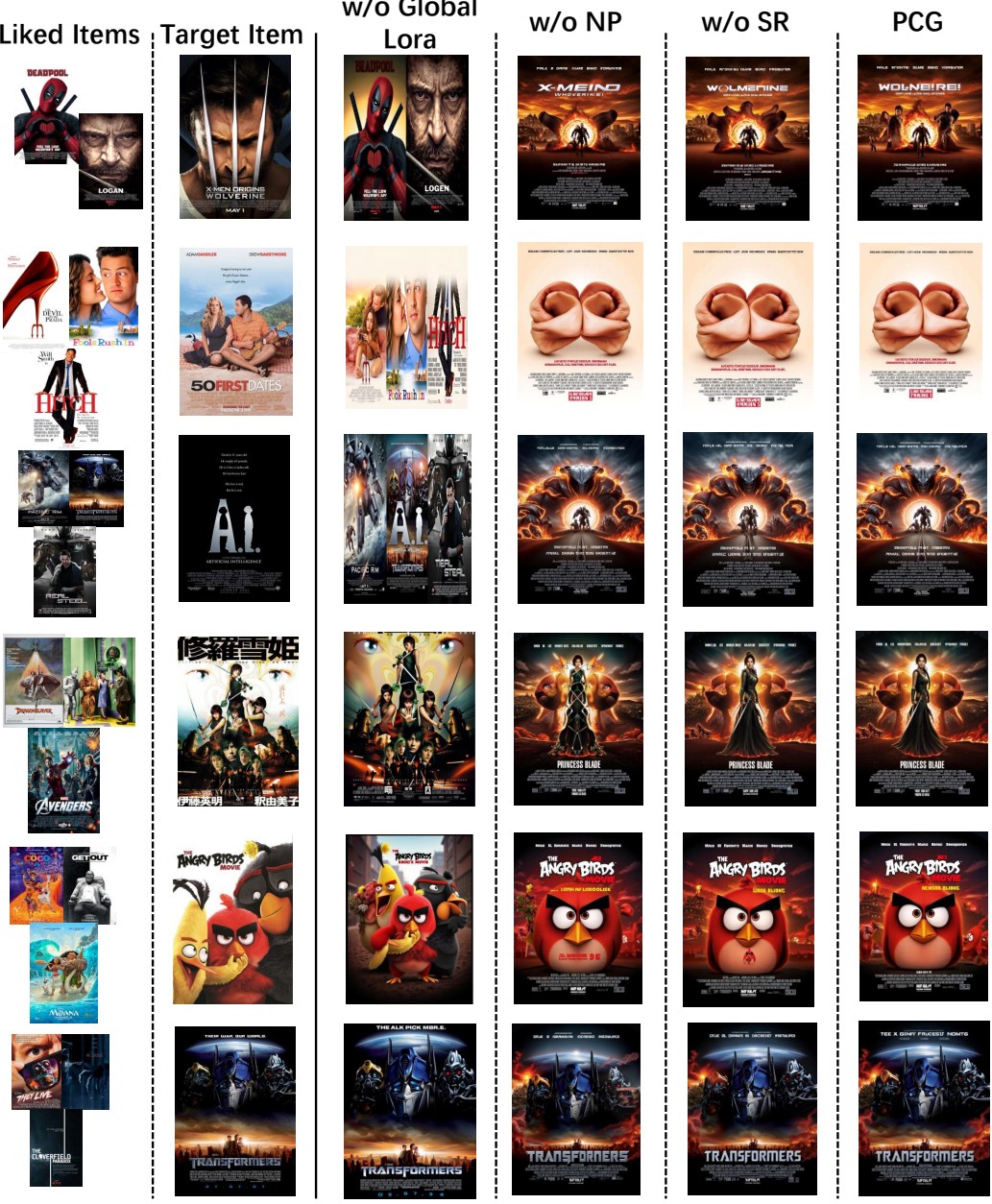

Figure 7: Ablation studies. Here NP represents Negative Prompt, and SR represents Score Reweighting.

# G    Implementation Details

As mentioned in Section H, we use Qwen3-8B [4] as the LLM to generate user inclinations and GPT-4o for evaluation. Here, we provide a detailed description of the configuration used for Qwen model. We generate outputs with the following parameters: a maximum of 128 new tokens, sampling enabled with a temperature of 0.7, top-$p$ sampling with a probability of 0.8, top-$k$ sampling with a limit of 20, and a minimum probability of 0.0.

When fine-tuning PCG LoRA based on `EasyControl`, we strictly follow its recommended settings. The overall learning rate is set to $1 \times 10^{-4}$ (based on the FLUX.1-dev pre-trained model, using a single A100-80G GPU with a batch size of 1). The two types of LoRAs in the two-stage training share the same learning rate.

The optimizer used is AdamW with the parameters:

$$\beta_1 = 0.9, \quad \beta_2 = 0.999, \quad \text{weight decay} = 1 \times 10^{-4}.$$

The dimension of the low-rank matrices is set to 128.

# H    Inclinations Analyzer Prompt

```
Role: You are an expert. Please extract the user's movie-related
    common styles, colors, types words and their probabilities from
    the following dialogue. Output format: a list, each element is [
    descriptive word, probability], probability is a float between 0
    and 1. Note that the descriptive word should NOT be the movie
    names. For example, a user liked movie Forrest Gump, the output
    should be like this: [['encourage', 0.8], ['comedy', 0.3]].

Dialogue content: {dialogue_text}

Role: You are an expert. Please extract the user's disliked movie-
    related common styles, colors, types words and their probabilities
     from the following dialogue. Output format: Some words that the
    user dislikes, each word is a string. If the user doesn't dislike
    any movie, please return an empty list. For example, a user
    disliked movie Saving Private Ryan, the output may be like this:
    war, horror.

Dialogue content: {dialogue_text}
```

Listing 1: Few-shot prompt template for inclinations analysis.

# I    GPT-4o Evaluation Prompt

```
Role: I first provide you some historical liked images and the target
    item image. Then I will provide you with a set of movie posters.
    Please evaluate them based on the following four dimensions, with
    each dimension being rated on a scale of 1 to 5. Be objective and
    fair in your analysis and provide clear reasoning for each rating.
     Finally, calculate the total score and rank the posters
    accordingly. The evaluation dimensions are as follows:

1. Visual Aesthetics
Are the color combinations harmonious and visually impactful?

Is the composition balanced and layered?

Does it have the quality and feel of a cinematic movie poster?
```

---

[4] https://huggingface.co/Qwen/Qwen3-8B

```
2   . Integration Level
Are the elements from the Target Item and History Liked Items
    naturally integrated? Like the colors, elements and roles of
    History Liked Items.

Does it appear seamless and unified, as if they belong to the same
    universe?

3   . Thematic Coherence
Does it successfully convey the main storyline of the Target Item
    while highlighting the characteristics of the Liked Items?

Is the background story immediately understandable to the viewer?

4   . Detail Representation
Are the background, characters, lighting, and text details well-
    refined?

Are there any obvious signs of poor blending or noticeable seams?

Scoring Criteria:
1: Very Poor - Completely uncoordinated

2: Poor - Noticeable flaws

3: Fair - Basic integration but lacks refinement

4: Good - Visually consistent and well-detailed

5: Excellent - Perfectly blended with cinematic quality
```

Listing 2: Prompt for GPT Evaluation.

## J   Original Conversations

```
[
  {
    "assistant": "Hi",
    "user": "Hello! How are you? Could you recommend some comedy's? I'
    ve had a tough day."
  },
  {
    "assistant": "I recommend [Meet the Parents (2000)], since  i t s
    a funny comedy and it can make you laugh. It tells the story of
    Greg, a future son-in-law and male nurse, who meets his strict
    father-in-law."
  },
  {
    "user": "No, I haven't."
  },
  {
    "assistant": "You have to see it. If you're interested in more
    family comedies, also try [Meet the Fockers (2004)]  the
    hilarious sequel. Another great option is [The Hangover (2009)],
    where three buddies wake up from a bachelor party in Las Vegas
    with no memory and the groom missing."
  },
  {
    "user": "I will check them out."
  },
  {
    "user": "Oh, [The Hangover (2009)] was funny"
```

```
    },
    {
      "assistant": "I recommend [Superbad (2007)], since it's hilarious
      and funny. It's about three high schoolers trying to bring alcohol
       to a party to impress their peers. I'm sure after watching, you
      can't stop laughing."
    },
    {
      "user": "Have not seen that either. Will have to check it out."
    },
    {
      "user": "Thank you for the recommendations"
    },
    {
      "assistant": "You're welcome, take care"
    }
]
```

Listing 3: Conversation for Row-3

```
[
    {
      "user": "Hey!",
      "assistant": "Hi!"
    },
    {
      "user": "I am in need of some good action movies, not superhero
      movies though or [The Fast and the Furious] movies. I love movies
      like [Speed (1994)] with Keanu Reeves, or [The Taking of Pelham
      123 (2009)] with Joh Travolta"
    },
    {
      "user": "Any suggestions?"
    },
    {
      "assistant": "Since you like action movies, then I think this
      movie will suit your taste: [Looper (2012)]. The sci-fi is not
      foolproof as the director would like it to be, but the film is
      appealing and entertaining enough with a nice blend of action and
      thrills. I believe that you will enjoy yourself while watching it
      ."
    },
    {
      "user": "I have with Bruce Willis! That one is such a great movie
      !"
    },
    {
      "assistant": "If you're looking for great action movies, I
      recommend [John Wick (2014)]. It's about a first-class killer who
      is forced to go back to work to get revenge after his retirement.
      It's one of Keanu's best action movies, so you should really try
      it!"
    },
    {
      "user": "I've heard of it and I know the sequel came out, but have
       never seen it"
    },
    {
      "user": "What is that about?"
    },
    {
      "assistant": "As you have watched [Looper (2012)], then you can
      try [John Wick (2014)]. It's one of Keanu's best! It was directed
      by Chad Stahelski, David Leitch. Other actors include Michael
```

```
        Nyqvist, Alfie Allen and Willem Dafoe. I believe this movie will
        give you an unprecedented experience. It's a similar storyline to
        [Taken (2008)]. A man seeking vengeance for his loved one. It's
        really good."
    },
    {
        "user": "That one sounds really good!"
    },
    {
        "user": "I think the story line is really good thank you"
    },
    {
        "assistant": "You're welcome!"
    },
    {
        "assistant": "I hope you check it out! Enjoy your day!"
    }
]
```

Listing 4: Conversation for Row-4

