# OpenReview forum: "Personalized Visual Content Generation in Conversational Systems"
_NeurIPS.cc/2025/Conference — NeurIPS 2025 poster_

### Official Review · Reviewer_Q8gb · 2025-06-28

**Clarity:** 3
**Significance:** 3
**Originality:** 3
**Rating:** 4
**Confidence:** 3

**Summary:**

This paper introduces PiViC generation, a unified framework for personalizing item images within conversational systems, addressing the "one-size-fits-all" dilemma in such systems. It leverages the strong language understanding capabilities of large models to efficiently infer user preferences, which are then used to construct personalized prompts that guide the image generation process.

**Questions:**

I am particularly concerned about Weakness 1, as it has a direct impact on the practical application of this approach.

**Ethical Concerns:**

["NO or VERY MINOR ethics concerns only"]

**Final Justification:**

I find the idea interesting, and my concerns have been addressed. Therefore, I am inclined to give a positive rating.

**Limitations:**

Yes

**Quality:**

3

**Strengths And Weaknesses:**

Strength：
1.  The authors propose a novel task of integrating generative models into recommendation systems, which is a highly innovative idea.
2.  The manuscript is clearly written and easy to understand.


Weaknesses：
1. I find your idea of combining diffusion models with the user’s preference to generate personalized content very interesting. However, I am curious about the practical application of this approach. In recommendation systems, the recommended items actually exist, whereas the content generated by diffusion models may not correspond to real items. For example, as shown in Figure 1, a movie poster can be generated based on a user's preferences, but the corresponding movie may not exist in reality. How do you envision this technology being applied in practice, given that the generated content might not have a real-world counterpart?
2. I think that both EasyControl-SUBJ and EasyControl-BG are components of PiViC. Therefore, it would be more appropriate to include a comparison with image-to-image methods that are not based on EasyControl.
3. I think it is difficult to observe the superiority of the proposed method in Figure 3. Besides, conducting a comprehensive ablation study is crucial because it reveals the effectiveness of each individual component.
4. In terms of methodology, I think the authors first leverage LLMs to construct text prompts, and then refer to EasyControl during the diffusion generation process. Therefore, the methodological novelty is somewhat limited.

---

> ### Author Rebuttal · Authors · 2025-07-31
>
> We sincerely thank you for the overall positive evaluation and rating of our work !!! It is our honor to discuss our research with you~~
>
> ---
>
> **W1.**
> Your concern directly aligns with one of our optimization goals — the **fidelity** of generated content. If the generated image deviates too far from the original target item, the meaning of personalization diminishes.
> As shown in our results (e.g., *Fig. 1* — the first row “Waterboy” and the last row “Taken”), the generated outputs match the target items well, and objective metrics also confirm PiViC’s strong fidelity performance.
>
> That said, due to the **inherent limitations of current generative models**, achieving complete fidelity remains challenging. We would like to emphasize that PiViC is a **complete personalization pipeline** — if the base model itself achieves perfect fidelity, our method can also deliver fully faithful personalization on top of it. Since fidelity is a fundamental challenge in image generation, we believe this is an area where *we, other researchers, and the broader community* must work together.
>
> ---
>
> **W2.**
> We have added an additional **image-to-image** baseline (Kandinsky) for reference:
>
> | Model          | History                                     |       |        |         |        | Target         |         | FID↓   |
> |----------------|----------------------------------------------|-------|--------|---------|--------|----------------|---------|--------|
> |                | CLIP-T↑     | CLIP-I↑ | LPIPS↓ | MS-SSIM↑ | DIS↑   | CLIP-T↑       | MS-SSIM↑ |        |
> | Kandinsky         | **26.93**       | 56.71    | 0.6940 | 0.0924     | 0.8120 | 29.99      | **0.3884**   | 25.24  |
> | PiViC (Ours)   | 26.52   | **59.12** | **0.6714** | **0.1589**   | **0.8336** | **30.52**         | 0.1938   | **21.88** |
>
> PiViC achieves **superior performance** in both personalization and fidelity.
>
> We also provide comparisons with **closed-source products** (GPT‑4o, Gemini 2.5‑Pro):
>
> | Model          | History                                     |       |        |         |        | Target         |         | FID↓   |
> |----------------|----------------------------------------------|-------|--------|---------|--------|----------------|---------|--------|
> |                | CLIP-T↑     | CLIP-I↑ | LPIPS↓ | MS-SSIM↑ | DIS↑   | CLIP-T↑       | MS-SSIM↑ |        |
> | GPT-4o         | 25.18       | 57.80    | 0.6857 | **0.1800**     | 0.8289 | **30.63**      | **0.2158**   | 23.28  |
> | Gemini 2.5-Pro | 25.06       | 58.98   | 0.6860  | 0.1426|        0.8229 | 28.86         | 0.1511 | 22.12  |
> | PiViC (Ours)   | **26.52**   | **59.12** | **0.6714** | 0.1589   | **0.8336** | 30.52         | 0.1938   | **21.88** |
>
> PiViC offers a **better trade-off between personalization and fidelity**, with a clear improvement in integrating *historical liked items*.
>
> ---
> **W3.**
> Indeed, as shown in our examples, **Superbad** (third row) integrates more background elements from the user’s historical styles (in softer tones), **Taken** (last row) combines the protagonist’s professional and composed demeanor with the forward-facing pose found in historical liked items, and **Waterboy** (first row) incorporates distinctive background motifs from historical inclinations. These examples illustrate PiViC’s ability to not only preserve fidelity to the target item but also seamlessly embed personalized elements drawn from historical interests.
>
> ---
> **W4.**
> Thank you for the suggestion. As an **application-driven** work, our primary contribution is proposing a **novel pipeline** to address the problem of monotonous content presentation in conversational systems.
> Our goal is to present the pipeline as a **holistic solution** rather than focusing solely on model-architecture-level innovations. Given that current generative systems are heavily dependent on pretrained base models, purely architectural changes may not be accessible or impactful for all researchers.
>
> From a pragmatic perspective, we aim for PiViC to serve as a **starting point** for personalized visual generation in conversational systems — inspiring more **innovative, competitive, and groundbreaking** work built upon it, and contributing fresh ideas to the community.

---

> > ### Comment · Reviewer_Q8gb · 2025-08-05
> >
> > I have reading this. Thank you for the clarification.

---

> ### Author Response · Authors · 2025-08-05
> **Thank you for your insightful review !**
>
> We sincerely appreciate your **positive rating** of our work, as well as the encouragement from the other reviewers. Your feedback has been a tremendous source of motivation. We look forward to building on these insights to further refine GenAI so that it can better serve users’ individual needs. Wishing you continued success in your work and good health.
>
> Best regards,
>
> The Authors

---

### Official Review · Reviewer_3nXA · 2025-06-30

**Clarity:** 3
**Significance:** 3
**Originality:** 3
**Rating:** 5
**Confidence:** 4

**Summary:**

This paper explores the tasks and new methods of personalized generation in conversational systems. This paper attempts to solve the problem that the content generated by conversational systems is not personalized enough, and proposes PiViC: using a large model to extract user inclinations, and using a flux architecture with multi-loras to incorporate the user's historical interests, so that the generated content is more in line with the user's preferences. Comprehensive experiments prove the effectiveness of the method.

**Questions:**

1. Did the author try more datasets? Although you mentioned that the current conversational recommendation system mainly uses these two movie datasets, is this a common practice or best practice? It would be better if there were data from more fields.
2. The article did not provide a more detailed interpretation of the case study of the generated results, for example, it could explain how the generated results better reflect the user's conversational interests.

**Ethical Concerns:**

["NO or VERY MINOR ethics concerns only"]

**Limitations:**

Yes

**Quality:**

3

**Strengths And Weaknesses:**

Strengths:
1. The motivation of the article is reasonable. For dialogue systems, previous work rarely considers content personalization and adaptation. This is an interesting idea.
2. The framework is innovative and solves the gap between user dialogue history content and visual content generation, so that the dialogue system can generate & display content that users like.
3. The experiment is detailed. The experiment covers modules such as subjective, objective, GPT analysis, ablation studies, and hyperparameter analysis, proving that the proposed PiViC framework can better reflect user historical interests.
4. The technology used is theoretically analyzed, which highlights the rationality of the Inclinations Analyzer and Visual Content Condition related modules the paper used.

Weaknesses:
1. Although the method is solid, the article mainly relies on the existing architecture. If the architecture can be designed to be more unified (for example, personalized generation can be performed without two stages), the contribution will be more highlighted.
2. The article did not conduct a human study. Does using GPT for scoring mean that the generated content is more in line with the user's interests and preferences?
3. Some major experiments (such as ablation studies) should be placed in the main text to ensure that the content is self-consistent.

---

> ### Author Rebuttal · Authors · 2025-07-31
>
> We greatly appreciate your high recognition of our work — it motivates us to continue improving. We are happy to address your questions.
>
> ---
>
> **W1.**
> This is indeed the direction we aim to explore next. However, as you can see, many current unified multimodal understanding–generation models (e.g., **Janus** [1]) still decouple *multimodal understanding* and *image generation* (even if they reside within the same model). We expect and believe that future designs will build upon our work with more integrated approaches.
>
> [1] Janus: Decoupling Visual Encoding for Unified Multimodal Understanding and Generation
>
> ---
>
> **W2.**
> We have attempted such experiments, but we found that participants in the human study often scored images according to their pre-existing personal preferences, which could bias the evaluation.
> Therefore, we ultimately chose to use **objective metrics** and **GPT-based evaluation** to assess generation quality.
>
> ---
>
> **W3.**
> Thank you for the suggestion. Other reviewers also mentioned this point. In the final version, we will include more important experiments — including the ablation study — directly in the main text.
>
> ---
>
> **Q1.**
> Our evaluation focuses on movie datasets because our work is designed for **conversational systems**, and to date, almost all such systems are **built on two movie datasets** [1–5], without adoption of other domains.
>
> [1] *Parameter-Efficient Conversational Recommender System as a Language Processing Task*, EACL’24
> [2] *Multi-Type Context-Aware Conversational Recommender Systems via Mixture-of-Experts*, Arxiv’25
> [3] *Towards Unified Conversational Recommender Systems via Knowledge-Enhanced Prompt Learning*, KDD’22
> [4] *COLA: Improving Conversational Recommender Systems by Collaborative Augmentation*, AAAI’23
> [5] *A Large Language Model Enhanced Conversational Recommender System*, Arxiv’23
>
> After a more careful investigation, we found that other niche conversational recommender datasets **do not** provide both *historical* and *target* item images, which makes them fundamentally incompatible with PiViC’s setting. We hope this clarifies our choice.
>
> ---
>
> **Q2.**
> We have provided more direct analyses of generated results in the **Supplementary Material.zip** (Appendix). Please refer to it for details. In the final version, we will also consider adding more intuitive qualitative comparisons of generated results.

---

> > ### Comment · Reviewer_3nXA · 2025-08-07
> > **Response to the authors**
> >
> > Thank you for your explanation. I'd like to maintain my current positive score.

---

### Official Review · Reviewer_Xfff · 2025-07-01

**Clarity:** 3
**Significance:** 2
**Originality:** 3
**Rating:** 4
**Confidence:** 4

**Summary:**

This paper introduces PiViC, a novel framework for personalized visual content generation in conversational systems, which leverages user inclinations extracted via LLMs and integrates them through a dual-stage LoRA mechanism (global and local) to adaptively guide image synthesis. Extensive experiments on benchmark datasets show that PiViC effectively balances fidelity to target items and personalization from user history, outperforming strong baselines like FLUX and EasyControl.

**Questions:**

1. Have you considered interactive or real-time updates to the personalized visual generation if a user changes preferences mid-dialog?
2. In your framework, how scalable is the dual-stage LoRA approach when conditioning on a large number of historical items?

**Ethical Concerns:**

["NO or VERY MINOR ethics concerns only"]

**Final Justification:**

The experimental results provided by the authors during the rebuttal stage have largely addressed my concerns, so I have increased my score accordingly.

**Limitations:**

1. The paper primarily benchmarks against tailored diffusion-based pipelines and does not include direct comparisons with recent large multimodal models (e.g., GPT-4o or Gemini) that can jointly handle long conversational histories and multi-image conditioning, which might offer strong baselines.
2. While PiViC effectively captures user inclinations through dialogue, it heavily relies on conversation-derived preferences and does not explore integrating richer user behavior signals (such as clicks or ratings), potentially limiting its personalization depth in broader applications.

**Quality:**

2

**Strengths And Weaknesses:**

### strengths
1. The paper is well-written and clearly structured, making easy to understand.
2. The paper demonstrates a thoughtful integration of language understanding and visual generation, enabling a seamless blend of user preferences with item-specific content.

---

### weaknesses
1. Lack of comparisons with alternative baselines, such as directly employing LVLMs for inclination analysis.
2. The experimental comparisons are not comprehensive; to my knowledge, models like GPT-4o and Gemini can process multiple images and long contexts. It would be valuable to see how such models perform on the E-ReDial and Inspired datasets.

---

> ### Author Rebuttal · Authors · 2025-07-31
>
> We sincerely thank you for reviewing our paper and for your positive recognition of its clarity and novelty. We are pleased to address your questions and hope this will persuade you to **raise your rating** for our work~~
>
> **W1.**
> Your suggestion is excellent. In the initial design, we did attempt to use VLMs as an analyzer to obtain users’ inclinations (by feeding both the dialogue and the mentioned movie poster images into the model) to achieve more stable user modeling. However, this approach required significantly more resources and inference time compared to PiViC, making it impractical.
> Nevertheless, we have supplemented our work with a comparison experiment using another open-source generative model (**Kandinsky**) on the E-Redial dataset as an alternative baseline:
>
> | Model          | History                                     |       |        |         |        | Target         |         | FID↓   |
> |----------------|----------------------------------------------|-------|--------|---------|--------|----------------|---------|--------|
> |                | CLIP-T↑     | CLIP-I↑ | LPIPS↓ | MS-SSIM↑ | DIS↑   | CLIP-T↑       | MS-SSIM↑ |        |
> | Kandinsky      | **26.93**   | 56.71    | 0.6940 | 0.0924     | 0.8120 | 29.99         | **0.3884** | 25.24  |
> | PiViC (Ours)   | 26.52       | **59.12** | **0.6714** | **0.1589** | **0.8336** | **30.52** | 0.1938   | **21.88** |
>
> This further demonstrates PiViC’s effectiveness.
>
> ---
>
> **W2 & L1.**
> Your proposed experiment is indeed *very nice*, and we are happy to accept it.
> Since GPT-4o and Gemini cannot generate images via API, we used the official GPT-4o platform and the Gemini 2.5 Pro subscription version to generate images. As you know, this **collection process is highly time-consuming and labor-intensive**, so we conducted the experiment on the E-Redial dataset.
> The prompt used for both GPT and Gemini was:
> ```
> Generate a movie poster of xxx based on the user interest shown in the below dialogue: \n <dialogue>
> ```
>
> Due to platform restrictions, some images could not be generated (likely because of prohibited content), with reminders such as:
>
> > I can't generate that image because the request violates our content policies.
> > Please feel free to share a different idea or request, and I'd be happy to help!
>
> This resulted in *11 test set images* for GPT-4o being excluded from evaluation. As closed-source models, both have been fully trained on target items, so we focus on their ability to integrate historical interests.
> Due to this year’s **rebuttal policy**, we cannot share generated images, but we believe the objective metrics below can **largely address your concerns**:
>
> | Model          | History                                     |       |        |         |        | Target         |         | FID↓   |
> |----------------|----------------------------------------------|-------|--------|---------|--------|----------------|---------|--------|
> |                | CLIP-T↑     | CLIP-I↑ | LPIPS↓ | MS-SSIM↑ | DIS↑   | CLIP-T↑       | MS-SSIM↑ |        |
> | GPT-4o         | 25.18       | 57.80    | 0.6857 | **0.1800**     | 0.8289 | **30.63** | **0.2158** | 23.28  |
> | Gemini 2.5-Pro | 25.06       | 58.98   | 0.6860  | 0.1426         | 0.8229 | 28.86     | 0.1511   | 22.12  |
> | PiViC (Ours)   | **26.52**   | **59.12** | **0.6714** | 0.1589     | **0.8336** | 30.52     | 0.1938   | **21.88** |
>
> We can see that while GPT-4o produces highly faithful posters, its ability to incorporate historical information is inferior to our proposed PiViC. This suggests these models have limited capacity for understanding and integrating conversational information.
> Gemini’s fidelity and personalization are weaker than GPT-4o’s, a finding consistent with reports from some existing benchmark sites. These results further **highlight PiViC’s strength in reflecting user interests in conversation**.
>
> ---
>
> **Q1.**
> This is an excellent question. Although we have not observed this phenomenon in our existing datasets, it is a valuable point.
> For such cases, an *inclinations register* can be maintained to store users’ long-term interests. When a shift occurs, both the stored inclinations and new ones can be analyzed by the analyzer, reducing the analysis length and allowing quick decoding of updated inclinations.
> Moreover, since LoRAs learn the ability to integrate historical liked items, they can still adapt when new inclinations appear.
>
> ---
>
> **Q2.**
> Scalability involves two aspects:
> (1) Analyzing inclinations in long dialogues, and
> (2) Handling a huge number of historical items.
>
> For long dialogues, our Q1 solution applies.
> For the dual-stage LoRA (handling historical & target images), the target image is always a single one, and regardless of the number of historical images, we concatenate them before conditioning. This extra cost lies only in the concatenation step (no extra computational overhead) and is negligible. Therefore, PiViC is scalable.
>
> ---
>
> **L2.**
> We share your concern. We tried to find a conversational dataset with explicit feedback (e.g., ratings or clicks), but due to dataset limitations, we only obtained dialogue-based signals.
> If you could point us to a conversational recommender dataset with direct supervision signals like clicks/ratings, it would indeed highlight the merit of our PiViC. We also plan to explore this direction and will incorporate click/rating signals in future versions. In fact, click/rating data could directly score inclinations, serving as a highly reliable reflection of user interests.
>
> Sincere thanks for your suggestion! : )

---

> > ### Author Response · Authors · 2025-08-05
> >
> > Dear Reviewer,
> >
> > As you can see, we have **added the relevant experiments** as requested. We wonder if they have addressed your concerns? This message is not intended to rush you — rather, we just wanted to reach out as the discussion phase is drawing to a close. If there is anything that might be holding you back from raising the score, please *feel free to let us know*. We'd love to reserve some time to further discuss and improve the work together~

---

> > > ### Comment · Reviewer_Xfff · 2025-08-05
> > >
> > > I sincerely appreciate the authors' efforts. I'm pleased to see the updated results, which largely address my concerns. Based on the improvements, I will revise my score, as I find the proposed framework to be both effective and well-aligned with its objectives.

---

### Official Review · Reviewer_kNWa · 2025-07-03

**Clarity:** 3
**Significance:** 3
**Originality:** 3
**Rating:** 4
**Confidence:** 3

**Summary:**

This paper introduces PiViC (Personalized Visual Content), a framework designed for generating personalized visual content in conversational systems. The authors propose two-stage LoRA, which combines Global LoRA for task-specific styles and Local LoRA for user-preferred visual elements, enabling the generation of images that reflect user preferences.

**Questions:**

1. I would like to suggest the authors performing broader evaluations on more benchmarks other than the movie conversational benchmarks, e.g., e-commerce product posters.

2. Considering this is an application technique, he authors should consider discussing computational complexities and wall-clock time measures more carefully.

3. I wonder the visual effects of adjusting the LoRA hyper-parameters. Provide more detailed analysis would be interesting.

**Ethical Concerns:**

["NO or VERY MINOR ethics concerns only"]

**Limitations:**

Yes.

**Paper Formatting Concerns:**

N/A.

**Quality:**

3

**Strengths And Weaknesses:**

- Stengths:

1. The paper is easy to follow. The figure illustrations are clear.
2. The proposed method is simple and extendable, and has potential industrial values.
3. The paper employs both objective metrics (e.g., FID, MS-SSIM) and subjective GPT-based evaluations to provide more thorough evaluation results.

- Weaknesses:

1. The evaluations are only performed on two movie conversational recommender benchmarks, which is relatively limited. Expanding the evaluation to a wider variety of domains (e.g., non-entertainment or non-image-based recommendations) would strengthen its generalizability.
2. The LoRAs and integration of multiple visual elements may increase computational cost, which could impact scalability, especially for real-time applications.
3. Ablation studies and analysis are limited.

---

> ### Author Rebuttal · Authors · 2025-07-31
>
> We are grateful for your recognition of our work, and we are very happy to answer your questions : )
>
> **W1 & Q1.**
> Our evaluation is conducted on movie datasets because our work targets *conversational recommender systems*, and so far, almost all conversational recommender systems are **built on two movie datasets** without using other datasets [1-5].
> [1] Parameter-Efficient Conversational Recommender System as a Language Processing Task, EACL'24
> [2] Multi-Type Context-Aware Conversational Recommender Systems via Mixture-of-Experts, Arxiv'25
> [3] Towards Unified Conversational Recommender Systems via Knowledge-Enhanced Prompt Learning, KDD'22
> [4] COLA: Improving conversational recommender systems by collaborative augmentation, AAAI'23
> [5] A Large Language Model Enhanced Conversational Recommender System, Arxiv'23
>
> Through a more detailed investigation, we found that other *niche* conversational recommender system datasets **do not** contain history and target item images, which creates a large gap from the PiViC setting. We hope you can understand this difference. Nevertheless, we have additionally included results on the E-Redial dataset, using *closed-source models* (GPT-4o, Gemini 2.5-Pro) for objective metric comparison, for your reference (in W2 & Q2).
>
> ---
>
> **W2 & Q2.**
> Using LoRAs indeed increases inference latency to some extent. However, considering real-time applications, we compared our method with mature commercial products (e.g., GPT-4o and Gemini 2.5-Pro):
>
> | Model   | Time per Image (s) |
> |---------|--------------------|
> | GPT-4o  | ~40s               |
> | Gemini  | ~25s               |
> | PiViC   | ~20s               |
>
> As shown, our runtime is within an acceptable range compared to the other two. In fact, we have already initiated new work to optimize the generation process via **KV cache**, since the same target item images are often used multiple times as keys and values during conditioning. Please stay tuned for further improvements!
>
> Additionally, we include the comparison of generation quality with these closed-source products:
>
> | Model          | History                                     |       |        |         |        | Target         |         | FID↓   |
> |----------------|----------------------------------------------|-------|--------|---------|--------|----------------|---------|--------|
> |                | CLIP-T↑     | CLIP-I↑ | LPIPS↓ | MS-SSIM↑ | DIS↑   | CLIP-T↑       | MS-SSIM↑ |        |
> | GPT-4o         | 25.18       | 57.80  | 0.6857 | **0.1800** | 0.8289 | **30.63**    | **0.2158** | 23.28  |
> | Gemini 2.5-Pro | 25.06       | 58.98  | 0.6860 | 0.1426     | 0.8229 | 28.86         | 0.1511   | 22.12  |
> | PiViC (Ours)   | **26.52**   | **59.12** | **0.6714** | 0.1589 | **0.8336** | 30.52         | 0.1938   | **21.88** |
>
> We can see that PiViC achieves a better balance between personalization and fidelity. GPT-4o generates highly faithful images, but still lags behind PiViC in personalization.
>
> ---
>
> **W3.**
> Thank you for your suggestion. Our ablation study is placed in the *Supplementary Material.zip* (Section J.2, to be moved into the main paper in the final version), which might be easy to overlook. The ablation study covers four aspects: **Sequence Invariance**, **Global LoRA**, **Negative Prompt**, and **Score Reweight**, along with objective metric comparisons. For the latter three, we also provide intuitive visual comparisons of the generated results.
>
> ---
>
> **Q3.**
> This experiment is indeed very interesting and highly valuable for understanding the internal mechanisms of LoRA. Unfortunately, due to this year's conference guidance, which **strictly forbids** authors from updating images in the repository, we cannot include them at this stage. However, we will add a discussion on this part in the final version.
>
> Thank you very much for your in-depth suggestions!

---

> > ### Comment · Reviewer_kNWa · 2025-08-09
> >
> > Thank you for your clarification. I would maintain my positive score as the authors has addressed most of my concerns.

---

### Author Response · Authors · 2025-08-09
**Thank you for your efforts!**

Dear SAC, AC, and the four reviewers,

We sincerely thank you for the time, effort, and valuable feedback you have dedicated to our work. We are delighted that **all reviewers have given us positive ratings**—this is an immense encouragement and motivation for us. Building upon this work as a foundation, we will strive to innovate further and deliver even more **solid** contributions in the future.

We acknowledge that generative models inherently have certain limitations. Therefore, in our future work, we will focus on refining *the generation of human-related details* and addressing the current issue where imperfect content fidelity may cause generated outputs to stray somewhat from the original intent. These improvements will enhance the practical value of *PiViC*.

We believe and look forward to seeing more excellent work **built on ours**, offering new solutions to these open problems, and further advancing personalization in AIGC for conversational systems.

Once again, we thank you for your efforts and wish you continued success in your research and careers.

Best regards,
The Authors of *PiViC*

---

### Decision · Program_Chairs · 2025-09-17

**Decision:**

Accept (poster)

**Comment:**

This paper introduces PiViC, a personalized visual content generation framework designed to personalize item images for conversational recommender systems. The authors develop an LLM-powered Inclinations Analyzer to analyze the user-system conversation history and identify users' likes and dislikes, generating personalized pictures based on this information. Global and local LoRA mechanisms are also proposed to learn historical context and maintain original image fidelity. The experiments are conducted on two movie recommender datasets, whose results are described in Tables 2 and 2.

Strengths:
+ The proposed method is simple, extensible, and has potential real-world applications
+ Thorough evaluations are conducted using both objective metrics and subjective GPT-based evaluations
+ The work is theoretically analyzed with the rational choice of the inclinations analyzer and visual content

Weaknesses:
- Using GPT as a judge can introduce bias in evaluating visual content, as shown in https://psrdataset.github.io/.

Overall, I agree with the reviewers' feedback and the authors' thoughtful rebuttal and support to see the paper accepted.